# Influencing factor of COVID-19 vaccination trust and hesitancy in Wonju city, South Korea

**Hocheol Lee[1], Eun Bi Noh[2], Ji Eon Kim[1], Juyeon Oh[3], Eun Woo Nam[1,2,4]***

**1** Yonsei Global Health Center, Yonsei University, Wonju, Gangwon-do, Republic of Korea, **2** Department of Health Administration, Yonsei University Graduate School, Wonju, Gangwon-do, Republic of Korea, **3** Department of Information Statistics, Yonsei University, Wonju, Gangwon-do, Republic of Korea, **4** Center of Evidence Based Medicine, Institute of Convergence Science, Yonsei University, Wonju, Gangwon-do, Republic of Korea

* ewnam@yonsei.ac.kr

**Data Availability Statement:** Data cannot be shared without permission by Wonju city hall (Healthy City team) publicly, South Korea. Data are available from the Wonju city hall Institutional Data Access (contact via Yeon-seok Jin,

## Abstract

Social capital (SC) has been documented to effectively reduce the spread of diseases, including COVID-19; however, research pertaining to SC and COVID-19 vaccination in Korea is lacking. This cross-sectional study conducted in the city of Wonju, Gangwon Province, Korea (n = 1,096) examined the differences in COVID-19 vaccine trust and hesitancy considering individual characteristics and investigated the effects of SC on COVID-19 vaccine trust and hesitancy. SC was measured based on 14 items pertaining to social trust, network, and norms. Responses regarding COVID-19 screening history, vaccine trust, and vaccine hesitancy were also assessed. SC scores did not differ between sexes, but differed significantly according to age and household income; thus, adults aged 70–79 years had the highest SC scores, and mean SC score increased significantly with income. COVID-19 vaccine trust differed significantly according to age, average household income, social organization involvement, and SC score. COVID-19 vaccine hesitancy differed significantly with age, SC score, and COVID-19 screening history. In univariate logistic regression, age, average household income, social organization involvement, and SC score were significant predictors of vaccine trust; in multivariable analysis, however, the identified predictors were age and SC. In particular, people with an SC score ≥50 were 2.660 times more likely to trust COVID-19 vaccines than those with lower scores. In multivariable analysis, age and SC were significant predictors of vaccine hesitancy. In particular, people with an SC score ≥50 were 1.400 times more likely not to be hesitant about receiving COVID-19 vaccines than people with lower scores. These results indicate that prioritizing policies to increase SC and trust in the government could boost the COVID-19 vaccination rate.

## Introduction

Since the first case of coronavirus disease 2019 (COVID-19) was reported in Korea on January 19, 2020, there have been a total of 24,740,635 cases and 28,364 deaths as of September 29, 2022 [1]. Following the World Health Organization (WHO) vaccination guidelines, Korea began COVID-19 vaccine rollout in February 2021—starting from healthcare workers and

jys12076@korea.kr) / Ethics Committee (contact via yuwirb@yonsei.ac.kr/ https://e-irb.yonsei.ac.kr/index.htm?) for researchers who meet the criteria for access to confidential data.

**Funding:** This research was supported by the Basic Science Research Program through the National Research Foundation of Korea (NRF) funded by the Ministry of Education (NRF-2021R1C1C2005464). The funders had no role in study design, data collection and analysis, decision to publish, or preparation of the manuscript.

**Competing interests:** The authors have declared that no competing interests exist.

moving on to older adults, epidemiology personnel, high-risk groups, and the general population—with a plan to achieve herd immunity by completely vaccinating 70% of the population by November 2021 [2]. The goal has been met, with an 86.3% vaccination rate as of September 2022.

In the Core Capacity Workbook for International Health Regulations, the World Health Organization (WHO) the following eight core capacities for the control and prevention of the spread of infectious diseases: a) National legislation, policy, and financing; b) Coordination and communications; c) Surveillance; d) Response; e) Preparedness; f) Risk communication; g) Human resources; and h) Laboratory [3]. Furthermore, it emphasized that governments should be equipped with adequate vaccine transport infrastructure, healthcare facilities, risk communication and monitoring systems, finances, and social capital (SC) to recommend COVID-19 vaccination [4].

SC was defined as "shared values in relation to connections and networks among individuals." It particularly encompasses social activities, trust, norms, and attachment [5, 6] and has been consistently documented as strongly associated with individuals' physical and mental health [7]. SC is also effective in alleviating the direct and indirect health threats posed by the COVID-19 pandemic. Previous studies in Korea and the United States have reported that regions with higher SC responded more proactively to the spread of COVID-19 and showed reduced spread of the disease [8–10],. In particular, groups with high trust in the government, a factor of SC, strictly adhered to COVID-19 guidelines for the good of the social network (community) and limited their daily movements in compliance with the government mandate [10].

The Korean government has initiated its COVID-19 vaccine rollout to achieve herd immunity by the second half of 2021. To this end, the government has striven to boost public trust in COVID-19 vaccines and encourage vaccination. Furthermore, research has consistently reported that vaccination is strongly correlated with income level [11]. While some studies have reported that SC positively impacts the decision to get vaccinated against COVID-19, research or evidence pertaining to SC and COVID-19 vaccination in Korea is lacking.

This study aimed to identify the influencing factors of COVID-19 vaccine trust and hesitancy, with two specific objectives: 1) to examine the differences in COVID-19 vaccine trust and hesitancy according to various participant characteristics and 2) to investigate the effects of SC on COVID-19 vaccine trust and hesitancy.

## Methods

### Study design

This cross-sectional study was conducted in the city of Wonju, Gangwon Province, Republic of Korea. The city of Wonju is divided into 14 *dong* (urban areas) and 9 *myeon* (rural areas). In 2020, its population was 357,710, of whom 290,157 were adults. In 2021, the city implemented a five-year health plan, which includes measures for COVID-19 vaccination. Hence, this study aims to investigate the association of COVID-19 vaccine trust and hesitancy with SC to present useful data for devising COVID-19 policies. The study population comprised Korean adult ($\geq$ 20 years) residents of Wonju.

### Study instrument

The study instrument was developed with a focus on SC, COVID-19 responses, and general characteristics. First, SC was measured based on 14 items pertaining to social trust (4 items), social network (5 items), and social norms (5 items), with each item rated on a 5-point Likert-type scale. Second, COVID-19-related responses were assessed using items developed in the

context of the study region and based on the annual Community Health Survey conducted by Statistics Korea. The questions asked were regarding COVID-19 screening history, COVID-19 vaccine trust, and COVID-19 vaccine hesitancy. The general characteristics assessed were sex, age, average household income, and area of residence. Ages were divided into 20s, 30s, 40s, 50s, and 60s. House income was classified as <1 million, 1–1.99 million, 2–2.99 million, 3–3.99 million, 4–4.99 million, and >5 million. This was based on the classification table of the Community Health Survey in Korea. Social organization involvement was divided into involved (1) and not involved (0) and included involvement in community or organizations, such as volunteering, young adult groups, older adult groups, married women groups, parent associations, sports clubs, self-governing bodies, and religious organizations.

## Data collection

Adult residents of Wonju aged 20 years or older were enrolled in this study. The study sample was extracted via probability proportional to size sampling to ensure that it was representative of the population. In Step 1, the sample size was allocated proportionately to age and population for each of the 23 urban and rural areas of WonjuIn Step 2, households were randomly selected from the list of households of a village within each sample stratified by age and sex, and a questionnaire survey was administered to these households. Using Raosoft, the minimum sample size required for a confidence level of 95% and significance of 5% was calculated at 1,197.

For the questionnaire survey, 18 enumerators were hired and trained from April 26 to May 3, 2022. A pilot survey was conducted in four regions on May 5, and the main survey was conducted from May 7 to May 17. The pilot survey was conducted on 100 participants in two urban and two rural regions, and reliability (Cronbach's alpha) was found to be .86. Further, the content and construct validities of the questionnaire were determined based on the results of the pilot survey.

A total of 1,248 participants completed the survey. After excluding those with careless responses (e.g. censored respondents) and those who withdrew from the survey, a total of 1,096 participants were included. In detail, 152 participants were identified as careless responders, including 113 who withdrew from the study (74.3%), 27 who mutilated the questionnaire (17.7%), and 12 who did not consent to participate in the final step of the study (7.8%). We thus analyzed 87.82% (1,096/1,248) of the collected data. We obtained informed consent in written form from all the respondents. They were also informed of their right to refuse to answer any question.

## Statistical analysis

We used the following statistical techniques to analyze the effects of SC on COVID-19 vaccine trust and hesitancy. First, SC according to each characteristic was visualized using box plots. Further, the differences in SC according to participants' characteristics were analyzed using t-tests. Second, the differences in COVID-19 vaccine trust and hesitancy according to general characteristics, social organization involvement, SC score, and COVID-19 infection history were analyzed using Pearson's chi-square tests. Third, predictors of COVID-19 vaccine trust and hesitancy were identified using binary logistic regression. All regression coefficients, odds ratios, t-values, and p-values were examined to determine the predictability of each factor of COVID-19 vaccine trust and hesitancy.

All statistical analyses were conducted using STATA 15(Stata Corporation, College Station, TX, USA), and the data were visualized using the R-4.11 (R Foundation for Statistical Computing, Vienna, Austria).

### Ethical considerations

All components of this survey were approved by the institutional review board (IRB) of Yonsei University (IRB document number: 1041849-202104-SB-063-02).

## Results

### Participant characteristics

There were more urban (72.4%) than rural dwellers (27.6%) and more men (53.5%) than women (46.5%) among the study participants. The most common age group was over 60 years (22.0%). A total of 31.0% of the participants were involved in a social organization, and 60.4% had an SC score of <50. A total of 23.3% of the participants had undergone COVID-19 screening, 30.2% trusted COVID-19 vaccines, and 62.2% were willing to be vaccinated against COVID-19. The most common household income was ≥ 5 million KRW (22.4%) (Table 1).

The visualization of SC scores based on participant characteristics using box plots showed that SC scores did not differ between sexes, but did differ significantly according to age, with adults aged 70–79 years having the highest SC scores ($p<0.001$). SC score also differed according to household income, with the mean SC score increasing significantly with increasing income ($p<0.001$). While the mean SC score was lower in the socially involved group, there were many outliers. Moreover, SC scores differed according to COVID-19 vaccine trust, with the group that did not trust COVID-19 vaccines having higher SC scores ($p<0.001$). Individuals with as opposed to without vaccine hesitancy showed significantly higher SC ($p < .001$). Urban dwellers displayed higher SC than rural dwellers but not to a significant extent ($p = .945$) (Fig 1).

### COVID-19 trust and hesitancy by participant characteristics

Next, the differences in COVID-19 vaccine trust and hesitancy according to participant characteristics were analyzed using Pearson's chi-squared tests. COVID-19 vaccine trust differed significantly according to age ($p<0.001$), average household income ($p = 0.040$), social organization involvement ($p = 0.001$), and SC score ($p<0.001$). COVID-19 vaccine hesitancy differed significantly according to age ($p < .001$), SC score ($p = 0.003$), and COVID-19 screening history ($p = 0.039$) (Table 2).

### Predictors of COVID-19 vaccine trust and hesitancy

Binary logistic regression analysis was performed to identify the predictors of COVID-19 vaccine trust and hesitancy. In the univariate analysis of vaccine trust, age, average household income, social organization involvement, and SC score were identified as significant predictors. In the multivariable analysis, age and SC score were identified as significant predictors. In particular, people with an SC score of ≥50 were 2.660 times more likely to trust COVID-19 vaccines than those with an SC score of <50 ($p<0.001$).

In multivariable logistic regression analysis of predictors of COVID-19 hesitancy, age and SC were identified as significant predictors. In particular, people with an SC score of ≥50 were 1.400 times more likely not to be hesitant about receiving COVID-19 vaccines than people with an SC score of <50 ($p<0.01$) (Table 3).

## Discussion

This study aimed to analyze the effects of SC on COVID-19 vaccine trust and hesitancy. Our results showed that SC scores did not differ between sexes but differed significantly according to age and increased with increasing household income. COVID-19 vaccine trust differed

**Table 1. Characteristics of respondents (n = 1,096).**

| | N | % |
|---|---|---|
| Area of residence | | |
| Urban | 793 | 72.4 |
| Rural | 303 | 27.6 |
| Sex | | |
| Male | 586 | 53.5 |
| Female | 510 | 46.5 |
| Age | | |
| 20–29 years | 224 | 20.4 |
| 30–39 years | 212 | 19.3 |
| 40–49 years | 206 | 18.8 |
| 50–59 years | 213 | 19.4 |
| $\geq$ 60years | 241 | 22.0 |
| Average household income | | |
| $\leq$ 1 million KRW | 113 | 103 |
| 1.00–1.99 million KRW | 136 | 12.4 |
| 2.00–2.99 million KRW | 202 | 18.4 |
| 3.00–3.99 million KRW | 233 | 21.3 |
| 4.00–4.99 million KRW | 166 | 15.1 |
| $\geq$ 5 million KRW | 246 | 22.4 |
| Social organization involvement | | |
| Yes | 340 | 31.0 |
| No | 756 | 69.0 |
| Social capital score | | |
| <50 | 662 | 60.4 |
| $\geq$50 | 434 | 39.6 |
| COVID-19 screening | | |
| Yes | 255 | 23.3 |
| No | 841 | 76.7 |
| COVID-19 vaccine trust | | |
| Yes | 331 | 30.2 |
| No | 765 | 69.8 |
| COVID-19 vaccine hesitancy | | |
| Not hesitant | 682 | 62.2 |
| Hesitant | 414 | 37.8 |

KRW: Korean won

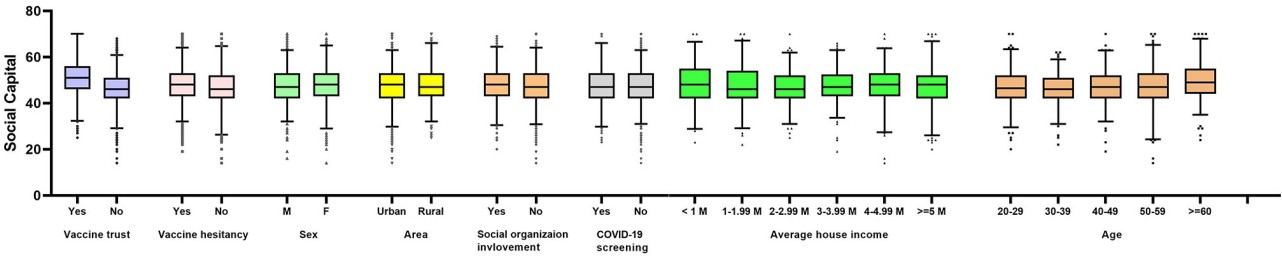

**Fig 1. Social capital by participant characteristics.**

**Table 2. Differences in COVID-19 vaccine trust and hesitancy.**

| | COVID-19 vaccine trust | | $\chi^2(p)$ | COVID-19 vaccine hesitancy | | $\chi^2(p)$ |
|---|---|---|---|---|---|---|
| | Yes | No | | Yes | No | |
| **Area of residence** | | | | | | |
| Urban | 299 (28.9%) | 564 (71.1%) | 2.382 (0.123) | 499 (63.1%) | 292 (36.9%) | 0.675 (0.411) |
| Rural | 102 (33.7%) | 201 (63.3%) | | 183 (60.4%) | 120 (39.6%) | |
| **Sex** | | | | | | |
| Male | 189 (32.3%) | 397 (67.7%) | 2.515 (0.113) | 363 (61.9%) | 223 (38.1%) | 0.084 (0.772) |
| Female | 142 (27.8%) | 368 (72.2%) | | 319 (62.8%) | 189 (37.2%) | |
| **Age** | | | | | | |
| 20–29 years | 39 (17.4%) | 185 (82.6%) | 58.881. (<0.001) | 117 (52.5%) | 106 (47.5%) | 35.972 (<0.001) |
| 30–39 years | 46 (21.7%) | 166 (78.3%) | | 111 (52.4%) | 101 (47.6%) | |
| 40–49 years | 59 (28.6%) | 147 (71.4%) | | 144 (69.9%) | 62 (30.1%) | |
| 50–59 years | 74 (34.7%) | 139 (65.3%) | | 142 (66.7%) | 71 (33.3%) | |
| ≥60 years | 113 (46.9%) | 128 (53.1%) | | 168 (70.0%) | 72 (30.0%) | |
| **Average household income** | | | | | | |
| ≤ 1 million KRW | 49 (43.4%) | 64 (56.6%) | 11.666 (0.040) | 69 (61.1%) | 44 (38.9%) | 3.876 (0.567) |
| 1.00–1.99 million KRW | 40 (29.4%) | 96 (70.6%) | | 78 (57.4%) | 58 (42.6%) | |
| 2.00–2.99 million KRW | 52 (25.7%) | 150 (74.3%) | | 125 (61.9%) | 77 (38.1%) | |
| 3.00–3.99 million KRW | 66 (28.3%) | 167 (71.7%) | | 143 (61.4%) | 90 (38.6%) | |
| 4.00–4.99 million KRW | 49 (29.5%) | 117 (70.5%) | | 112 (67.9%) | 53 (32.1%) | |
| ≥ 5 million KRW | 75 (30.5%) | 171 (69.5%) | | 155 (63.3%) | 90 (32.1%) | |
| **Social organization involvement** | | | | | | |
| Yes | 127 (27.0%) | 213 (73.0%) | 11.961 (0.001) | 221 (61.1%) | 119 (38.9%) | 1.487 (0.223) |
| No | 204 (37.4%) | 552 (62.6%) | | 461 (65.0%) | 293 (35.0%) | |
| **Social capital score** | | | | | | |
| <50 | 142 (21.5%) | 520 (78.5%) | 60.727 (<0.001) | 388 (56.9%) | 272 (41.2%) | 8.941 (0.003) |
| ≥50 | 189 (43.5%) | 245 (56.5%) | | 294 (67.7%) | 140 (32.3%) | |
| **COVID-19 screening** | | | | | | |
| Yes | 71 (30.9%) | 184 (69.1%) | 0.876 (0.349) | 145 (64.0%) | 110 (36.0%) | 4.249 (0.039) |
| No | 260 (27.8%) | 581 (72.2%) | | 537 (56.9%) | 302 (43.1%) | |

KRW: Korean won

significantly according to age, average household income, social organization involvement, and SC score. COVID-19 vaccine hesitancy differed significantly according to age, SC score, and COVID-19 screening history. In univariate logistic regression, age, average household income, social organization involvement, and SC score were found to be significant predictors of vaccine trust; in multivariable analysis, the identified predictors were age and SC. In multivariable analysis, age and SC were significant predictors of vaccine hesitancy.

Regarding the demographic characteristics of the participants, 72.4% were urban dwellers, which was similar to the reported percentage of urban dwellers (76.6%) for the city of Wonju [12], an urban-rural complex city. A total of 31.0% of the participants were involved in a social organization, which was higher than the rate (23.0%) reported in a previous study [13]. In our study, 30.2% of the participants stated that they trusted COVID-19 vaccines, which was lower than the global rate of vaccine trust (42.2%) [14]. Regarding COVID-19 vaccine hesitancy, 62.2% of the participants said they were not hesitant about being vaccinated, which was low considering that the average level of willingness to be vaccinated against COVID-19 worldwide is higher than 70.0% [15, 16]. The reasons for this low willingness may include the continuous

**Table 3. Results of logistic regression for predictors of COVID-19 vaccine trust and hesitancy.**

| Variable | COVID-19 vaccine trust | | | | COVID-19 vaccine hesitancy | | | |
|---|---|---|---|---|---|---|---|---|
| | cOR | 95% CI | aOR | 95% CI | cOR | 95% CI | aOR | 95% CI |
| Area of residence | | | | | | | | |
| Urban | Ref. | | Ref. | | Ref | | Ref. | |
| Rural | 1.250 | 0.941–1.659 | 1.185 | 0.874–1.606 | 0.892 | 0.680–1.171 | 0.858 | 0.648–1.135 |
| Sex | | | | | | | | |
| Male | Ref. | | Ref. | | Ref | | Ref. | |
| Female | .811 | 0.625–1.051 | .808 | 0.612–1.066 | 1.037 | 0.811–1.325 | 1.018 | 0.790–1.312 |
| Age | | | | | | | | |
| 20–29 years | Ref. | | Ref. | | Ref | | Ref. | |
| 30–39 years | 1.314 | 0.817–2.114 | 1.369 | 0.835–2.246 | 0.982 | 0.683–1.451 | 0.976 | 0.663–1.438 |
| 40–49 years | 1.904** | 1.203–3.012 | 2.000** | 1.230–3.230 | 2.104*** | 1.415–3.130 | 2.049** | 1.359–3.090 |
| 50–59 years | 2.525*** | 1.617–3.944 | 2.386*** | 1.488–3.8263.819 | 1.812** | 1.230–2.670 | 1.706* | 1.138–2.558 |
| ≥60 years | 4.188*** | 2.729–6.425 | 3.710**** | 2.323–5.923 | 2.114*** | 1.444–3.095 | 2.225*** | 1.469–3.370 |
| Average household income | | | | | | | | |
| ≤1 million KRW | Ref. | | Ref. | | Ref | | Ref. | |
| 1.00–1.99 million KRW | .544* | 0.322–0.919 | 0.643 | 0.366–1.128 | 0.858 | 0.516–1.426 | 0.573 | 0.573–1.636 |
| 2.00–2.99 million KRW | .453** | 0.278–0.737 | 0.628 | 0.370–1.067 | 1.035 | 0.645–1.661 | 1.337 | 0.812–2.202 |
| 3.00–3.99 million KRW | .516** | 0.323–0.825 | 0.15 | 0.432–1.237 | 1.013 | 0.639–1.607 | 1.217 | 0.746–1.984 |
| 4.00–4.99 million KRW | .547** | 0.332–0.902 | 0.687 | 0.410–1.149 | 1.348 | 0.818–2.221 | 1.553 | 0.911–2.647 |
| ≥5 million KRW | .573** | 0.361–0.908 | 0.841 | 0.410–0.245 | 1.098 | 0.694–1.737 | 1.383 | 0.845–2.262 |
| Social organization involvement | | | | | | | | |
| Yes | Ref. | | Ref. | | Ref | | Ref. | |
| No | 1.613*** | 1.229–2.118 | 1.065 | 0.790–1.4207 | 1.180 | 0.904–1.541 | .965 | 0.724–1.287 |
| COVID-19 screening | | | | | | | | |
| No | Ref. | | Ref. | | Ref | | Ref. | |
| Yes | .862 | 0.632–1.176 | 1.006 | 0.723–1.402 | 0.741* | 0.557–0.986 | 0.803 | 0.598–1.819 |
| Social capital score | | | | | | | | |
| <50 | Ref. | | Ref. | | Ref | | Ref. | |
| ≥50 | 2.825*** | 2.166–3.684 | 2.660*** | 2.018–3.504 | 1.472** | 1.142–1.898 | 1.400** | 1.077–1.819 |
| Hosmer–Lemeshow | | | .222 | | | | | .112 |

*$p<0.05$,

**$p<0.01$,

***$p<0.001$

KRW: Korean won, aOR: adjusted odds ratio

decline of the Korean government's approval ratings, limitations of risk communication, and spread of negative views due to the COVID-19 infodemic via the Internet and media [17]. A strong belief in inaccurate information about COVID-19 has been found to be influenced more by social media than the news [18], and considering that social media usage escalated by 87% during the COVID-19 pandemic compared to that in the pre-COVID-19 period, the younger generation—comprising the predominant users of social media—is likely to be vulnerable to the infodemic [19]. In addition to these factors, other personal factors such as education and income levels were found to have a strong influence on COVID-19 vaccine acceptance. By the same token, we speculate that the educational and income gaps that exist between the urban and rural regions of Wonju might have influenced vaccine acceptance [11].

SC has been reported to influence health, well-being, social support, sociability, and social standing; it has a particularly greater impact on older adults [20–22]. In our study sample, SC increased with age and increasing social involvement. One of the reasons for higher SC among older individuals is that older people engage in fewer economic activities, and social welfare policies are mostly focused on recommending and encouraging the social involvement of older adults [23].

The first objective of this study was to analyze the differences in COVID-19 vaccine trust and hesitancy according to the various characteristics of the study participants. COVID-19 vaccine trust was found to differ significantly according to age, household income, social involvement, and SC score, while COVID-19 vaccine hesitancy differed significantly according to age, SC score, and COVID-19 screening history. We confirmed that COVID-19 vaccine trust and acceptance increased significantly with advancing age, which is consistent with previous findings [24]. In particular, a recent telephonic survey of 1,200 people in Hong Kong showed that 42.2% of the respondents trusted COVID-19 vaccines and were willing to be vaccinated. By age, the COVID-19 vaccine acceptance rate was below 10% in the age group 16–54 years, but it increased with advancing age to 21.7% among those aged ≥55 years and 53.5% among those aged ≥65 years [24]. On the other hand, a Southeast Asian study reported that older adults are indeed vulnerable to COVID-19, but most of this population is retired and spends most of its time at home. Therefore, the study argued that the older adult population is at a lower risk of exposure to COVID-19 and, consequently, shows lower vaccine acceptance [25].

Increased COVID-19 vaccine trust and acceptance with advancing age is associated with the channels through which individuals acquire information and also the frequency of receiving information. Young individuals are frequently exposed to potentially inaccurate information through social media and are likely to encounter rumors and misinformation. In contrast, older adults generally obtain information about COVID-19 vaccines through public mass media, such as news outlets, and they are thus given relatively more credible information, which has probably contributed to boosting their COVID-19 vaccine trust [18]. Contrariwise, people who are provided with correct information have a detailed understanding of the risks of COVID-19 and, consequently, show high vaccine hesitancy [11]. However, as the WHO continually advocates the safety of COVID-19 vaccines based on literature evidence, the Korean government continues to recommend and promote vaccination through the media.

The second objective of this study was to identify the predictors of COVID-19 vaccine trust and hesitancy. Age and SC score were identified as predictors of COVID-19 vaccine trust and hesitancy. People with higher SC scores showed higher levels of COVID-19 vaccine trust and acceptance. SC comprises four dimensions—civic engagement, trust in the government, social belonging, and social trust—and a high SC score indicates that the individual trusts both the social community and the government and is actively involved in activities as a member of the society [26, 27]. Previous studies have reported that SC is effective in provoking behaviors that reduce COVID-19-related health risk [9], and that regions with greater SC deal with infectious diseases more proactively and thus have fewer cases in the community [7, 8]. SC-oriented public health responses can complement the limitations of top-down policies, and it is important to boost social solidarity and trust to achieve the public health goal of battling COVID-19 [28]. Thus, SC is expected to increase the COVID-19 vaccination rate and positively contribute to the efforts of communities to halt the spread of the disease. In particular, a previous study has reported that civic engagement and trust in the government had positive effects on COVID-19 responses, while trust in and a sense of belonging in affiliated groups had a negative effect on COVID-19 responses [29]. Another study reported that among the various elements of SC, low trust in institutions hindered COVID-19 response measures such as physical distancing [30].

Moreover, societies with high social trust may actually be more vulnerable to fake news about the severity of COVID-19, fake treatments, and criticisms against policies such as physical distancing [31].

This study has a few limitations. First, due to the cross-sectional study design, we could not identify the causative factors in a time-series analysis from 2020, when the index case of COVID-19 occurred in Korea. Second, although the city of Wonju, where this study was conducted, has a population of 300,000, the findings cannot be expected to be nationally representative. Subsequent studies should recruit participants from other and more diverse regions to ensure the generalizability of the findings in Korea.

## Conclusion

This study confirmed that SC is a predictor of COVID-19 vaccine trust and hesitancy, and that people with higher SC have greater trust in COVID-19 vaccines and are more willing to be vaccinated. The Korean government chose herd immunity as a key strategy for controlling COVID-19 and has implemented COVID-19 vaccination plans accordingly. Our results suggest that the government should strive to increase SC to boost the COVID-19 vaccination rate, which is crucial in pursuing long-term COVID-19 response measures to combat the prolonged pandemic. In particular, the government should prioritize policies to increase social involvement and trust in the government. To this end, the Korean government will need to transparently provide reliable COVID-19 information through various channels such as the media and the Internet through COVID-19 experts and trusted celebrities to increase its credibility.

## Supporting information

**S1 File.**
(PDF)

**S2 File.**
(PDF)

## Acknowledgments

We express our gratitude to everyone working to overcome COVID-19 in South Korea and throughout the world. Furthermore, we express our gratitude to the Healthy City team, Wonju City Hall.

## Author Contributions

**Conceptualization:** Hocheol Lee, Eun Woo Nam.

**Data curation:** Hocheol Lee, Eun Bi Noh, Juyeon Oh.

**Formal analysis:** Hocheol Lee.

**Funding acquisition:** Hocheol Lee.

**Methodology:** Hocheol Lee.

**Project administration:** Eun Woo Nam.

**Software:** Hocheol Lee, Juyeon Oh.

**Supervision:** Hocheol Lee, Eun Woo Nam.

**Visualization:** Eun Bi Noh.

**Writing – original draft:** Hocheol Lee, Eun Bi Noh, Ji Eon Kim, Eun Woo Nam.

**Writing – review & editing:** Hocheol Lee, Ji Eon Kim, Juyeon Oh, Eun Woo Nam.

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
