## [Decision Letter · Decision Letter 0]

20 Jun 2022

PONE-D-21-29174Association between social capital and COVID-19 vaccine trust and hesitancyPLOS ONE

Dear Dr. Nam,

Thank you for submitting your manuscript to PLOS ONE. After careful consideration, we feel that it has merit but does not fully meet PLOS ONE’s publication criteria as it currently stands. Therefore, we invite you to submit a revised version of the manuscript that addresses the points raised during the review process. Apologies for the delay, and thanks for your patience. After experiencing a great reluctance from many possible candidates (a sensitive topic)n, this paper has been reviewed by an acknowledged expert in this field. Overall, our referee highlights the interest raised by the manuscript, but also emphasizes on several technical & analytical shortcomings that must be amended by the authors before reconsidering this manuscript for publication. Therefore, the revisions required are basically major. Please find the comments below, at the end of this email.

We look forward to receiving your revised manuscript.

Kind regards,

Sergio A. Useche, Ph.D.

Academic Editor

PLOS ONE

Journal Requirements:

5. Thank you for stating the following financial disclosure: "This research was supported by the Basic Science Research Program through the National Research Foundation of Korea (NRF) funded by the Ministry of Education (NRF-2021R1C1C2005464)."

Please state what role the funders took in the study.  If the funders had no role, please state: "The funders had no role in study design, data collection and analysis, decision to publish, or preparation of the manuscript.

Reviewers' comments:

Reviewer's Responses to Questions

**Comments to the Author**

1. Is the manuscript technically sound, and do the data support the conclusions?

Reviewer #1: Yes

2. Has the statistical analysis been performed appropriately and rigorously? 

Reviewer #1: No

3. Have the authors made all data underlying the findings in their manuscript fully available?

Reviewer #1: No

4. Is the manuscript presented in an intelligible fashion and written in standard English?

Reviewer #1: Yes

5. Review Comments to the Author

Reviewer #1: I read the manuscript entited "Association between social capital and COVID-19 vaccine trust and hesitancy". The concept is interesting but already it is old idea. I see many article in online about the COVID-19 vaccine hesitency. However, most of them are missing in present manuscript. Therefore, authors need to improve introduction and discussion based on present published articles. See some specific comments bellow:

- consent from individual was missing. If you have that, add in the data colection section

-Table 1: Age categories should be minimized, so many catagories is meaningless. So I suggested to have only 3/4 categories of age.

-How you divided the household incomes? You need to add references in the method section, how you divided the income division? Accordingly, re-arrange the Table 1 based on income categories!

-Rearrange Table 2 and 3 based on re-arranged categories of age and household incomes.

-What is your key findings from your study? Need to explain key messages in the conclusion part. Based on the key message, what is your recommendations.

-In Figure 1, you need to adjust age, and household incomes.

6. PLOS authors have the option to publish the peer review history of their article (what does this mean?). If published, this will include your full peer review and any attached files.

Reviewer #1: No

---

## [Author Response · Author response to Decision Letter 0]

8 Jul 2022

We very appreciate you once again for receiving that comment by reviewers. 

We tried to revise/change by following the comments of reviewers. 

As result, we agreed this manuscript improved its quality due to comments.

We hope to hear a positive decision from Reviewers, Editorial Board and Editor. 

Thank you again.

Eun Woo Nam

Hocheol Lee

---

## [Decision Letter · Decision Letter 1]

1 Sep 2022

PONE-D-21-29174R1Association between social capital and COVID-19 vaccine trust and hesitancyPLOS ONE

Dear Dr. Nam,

Thank you for submitting your manuscript to PLOS ONE. After careful consideration, we feel that it has merit but does not fully meet PLOS ONE’s publication criteria as it currently stands. Therefore, we invite you to submit a revised version of the manuscript that addresses the points raised during the review process.

We look forward to receiving your revised manuscript.

Kind regards,

Harapan Harapan, MD, PhD

Academic Editor

PLOS ONE

Journal Requirements:

Additional Editor Comments:

Please revised your manuscript based on the previous suggestion of the reviewers. Fail to do so will result rejection. I will ask for next round review to the reviewers

Reviewers' comments:

Reviewer's Responses to Questions

**Comments to the Author**

1. If the authors have adequately addressed your comments raised in a previous round of review and you feel that this manuscript is now acceptable for publication, you may indicate that here to bypass the “Comments to the Author” section, enter your conflict of interest statement in the “Confidential to Editor” section, and submit your "Accept" recommendation.

Reviewer #1: (No Response)

2. Is the manuscript technically sound, and do the data support the conclusions?

Reviewer #1: Partly

3. Has the statistical analysis been performed appropriately and rigorously? 

Reviewer #1: N/A

4. Have the authors made all data underlying the findings in their manuscript fully available?

Reviewer #1: No

5. Is the manuscript presented in an intelligible fashion and written in standard English?

Reviewer #1: No

6. Review Comments to the Author

Reviewer #1: (No Response)

7. PLOS authors have the option to publish the peer review history of their article (what does this mean?). If published, this will include your full peer review and any attached files.

Reviewer #1: No

---

## [Author Response · Author response to Decision Letter 1]

4 Sep 2022

Dear Reviewers,

First, we are very appreciated for your helpful review. I discussed with all authors for your comments. And we revised/corrected the context as all manuscript. 

We attached the two manuscripts, 1) clean manuscript, 2) tracked manuscript. In the ’ 2) track manuscript’ file, our responses/revisions are highlighted using tracked system on word.

Thank you for review in advance.

Prof. Eun Woo Nam.

---

## [Decision Letter · Decision Letter 2]

19 Sep 2022

PONE-D-21-29174R2Association between social capital and COVID-19 vaccine trust and hesitancyPLOS ONE

Dear Dr. Nam,

Thank you for submitting your manuscript to PLOS ONE. After careful consideration, we feel that it has merit but does not fully meet PLOS ONE’s publication criteria as it currently stands. Therefore, we invite you to submit a revised version of the manuscript that addresses the points raised during the review process.

We look forward to receiving your revised manuscript.

Kind regards,

Harapan Harapan, MD, PhD

Academic Editor

PLOS ONE

Additional Editor Comments:

One of the reviewers attached the file. 

Reviewers' comments:

Reviewer's Responses to Questions

**Comments to the Author**

1. If the authors have adequately addressed your comments raised in a previous round of review and you feel that this manuscript is now acceptable for publication, you may indicate that here to bypass the “Comments to the Author” section, enter your conflict of interest statement in the “Confidential to Editor” section, and submit your "Accept" recommendation.

Reviewer #2: (No Response)

Reviewer #3: (No Response)

2. Is the manuscript technically sound, and do the data support the conclusions?

Reviewer #2: Partly

Reviewer #3: Yes

3. Has the statistical analysis been performed appropriately and rigorously? 

Reviewer #2: No

Reviewer #3: No

4. Have the authors made all data underlying the findings in their manuscript fully available?

Reviewer #2: No

Reviewer #3: No

5. Is the manuscript presented in an intelligible fashion and written in standard English?

Reviewer #2: No

Reviewer #3: Yes

6. Review Comments to the Author

Reviewer #2: Thank you for the opportunity of reviewing the manuscript whose objective of is to analyze the differences in COVID-19 vaccine trust and hesitancy according to the various participants’ characteristics, including the social capital. Please find below my comments:

Title

1. I’m pretty sure the study does not only cover the association between social capital and COVID-19 vaccine. Other factors are also studied. Please modify the title to be more representative of the content.

Introduction

1. Line 51. “eight core capacities” please elaborate.

2. Line 59. “SC has been fund to effectively” please check the accuracy

3. Line 60—61. “Previous studies have reported that regions with higher SC..” Where were the studies conducted? The elaboration is important because the submitted study has a novelty of its population being Koreans.

4. Is COVID-19 vaccination a mandatory in Korea? How is the progress so far?

5. To strengthen the background authors may incorporate the following studies revealing about the COVID-19 acceptance. And please focus on the detailed factors (such as income, safety, etc.).

Suggesting:

Sallam et al. Narra J 2022; 2(1): e74 – doi: 10.52225/narra.v2i1.74

Rosiello et al. Narra J 2021; 1(3): e55-doi: 10.52225/narra.v1i3.55

Methods

1. Calculation of sample size and sample randomization should be disclosed in details.

2. “..careless responses (e.g. censored respondents) and those who withdrew from the survey..” How could censored respondents are categorized as those giving careless responses? Why the participants withdrew from the survey? Didn’t they receive informed consent first?

3. Inclusion and exclusion criteria need to be explicitly disclosed. How authors determine the participants are Korean citizens?

4. Questionnaire validation should be explicitly disclosed. The company identity (name and address) for each statistical software should be stated.

5. What “social organization involvement” means? Is being a member is enough? Should the social organization officially registered? This has to be declared in the methods.

Results

1. Please pay attention on how the data are cited in the text. Usually, the paragraph begins with a sentence indicating where the data are presented. For instance, “Characteristics of the participants have been presented in Table 1”. If authors decide to cite the data in parenthesis, the citation should be made in the first and the last sentence where the data are discussed in the paragraph.

2. For age, consider years old instead of years only.

3. As per international standards, the writing of p<0.000 should be revised to p<0.001

Discussion

1. Authors argue the low willingness in younger generation was resulted from the disinformation spread through social medias, while also arguing that the older generation consumes more reliable information through traditional news sources. Authors cited reference no 6 to strengthen the claim. I don’t think ref 6 is appropriately cited in the manuscript, the literature mainly reports on the preference of news sources for COVID-19 information. Age was only considered as factor affecting the participants’ knowledge on COVID-19, not on their media preference. My suggestion is to modify this part of discussion for better judicious interpretation by readers. Other pronounced factors affecting the disinformation pertaining to COVID-19 (such as education) should be discussed instead.

2. Can you perform the analysis on “social media vs. age” and “social media vs. vaccine trust/hesitancy”?

3. Authors should know that the hesitancy is not always coming from the absence of the knowledge (because of social media as claimed in the manuscript). People who are well informed could also be hesitant because of the safety risk of the vaccine (Rosiello et al. Narra J 2021; 1(3): e55-doi: 10.52225/narra.v1i3.55). Please revise the discussion by considering this.

4. Is there any limitation on the data quality of the SC. For example, being biased or not able to confirm?

Conclusion

1. “COVID-19 experts and celebrities to increase its credibility” Why celebrities are included to increase the credibility?

Reviewer #3: Thank you for the opportunity to review this article. This article has been revised twice. However, I still detected some unclear issues related to this study. Therefore, I include some suggestions that I believe will improve the quality of the manuscript. Please encounter my specific comments in the attached file.

7. PLOS authors have the option to publish the peer review history of their article (what does this mean?). If published, this will include your full peer review and any attached files.

Reviewer #2: **Yes: **Muhammad Iqhrammullah

Reviewer #3: No

---

## [Author Response · Author response to Decision Letter 2]

13 Oct 2022

Dear Reviewer:

We appreciate your careful review of our manuscript. All authors have read and discussed your comments and revised/corrected the manuscript accordingly. 

Please find attached two manuscripts, one is a clean manuscript and the other is a tracked manuscript, which contains all our revisions made using the “Track Changes” feature in Microsoft Word.

We are grateful for your comments that have helped us improve the quality of our manuscript. Thank you in advance for reviewing our responses and revisions.

Sincerely,

Prof. Eun Woo Nam

---

## [Editor Report · Decision Letter 3]

17 Oct 2022

PONE-D-21-29174R3Influencing Factor of COVID-19 Vaccination trust and hesitancy in Wonju city, South KoreaPLOS ONE

Dear Dr. Nam,

Thank you for submitting your manuscript to PLOS ONE. After careful consideration, we feel that it has merit but does not fully meet PLOS ONE’s publication criteria as it currently stands. Therefore, we invite you to submit a revised version of the manuscript that addresses the points raised during the review process.

We look forward to receiving your revised manuscript.

Kind regards,

Harapan Harapan, MD, PhD

Academic Editor

PLOS ONE

Journal Requirements:

Additional Editor Comments (if provided):

Please delete the sample size calculation formula since authors did not explain this formula.
---

## [Editor Report · Decision Letter 4]

19 Oct 2022

Influencing Factor of COVID-19 Vaccination trust and hesitancy in Wonju city, South Korea

PONE-D-21-29174R4

Dear Dr. Nam,

We’re pleased to inform you that your manuscript has been judged scientifically suitable for publication and will be formally accepted for publication once it meets all outstanding technical requirements.

Kind regards,

Harapan Harapan, MD, PhD

Academic Editor

PLOS ONE
---

## [Editor Report · Acceptance letter]

25 Oct 2022

PONE-D-21-29174R4 

Influencing Factor of COVID-19 Vaccination trust and hesitancy in Wonju city, South Korea 

Dear Dr. Nam:

I'm pleased to inform you that your manuscript has been deemed suitable for publication in PLOS ONE. Congratulations! Your manuscript is now with our production department. 

Kind regards, 

on behalf of

Dr. Harapan Harapan 

Academic Editor

PLOS ONE